# The Multienzyme Complex Nature of Dehydroepiandrosterone Sulfate Biosynthesis

**DOI:** 10.3390/ijms25042072

**Published:** 2024-02-08

**Authors:** Anastasiya Tumilovich, Evgeniy Yablokov, Yuri Mezentsev, Pavel Ershov, Viktoriia Basina, Oksana Gnedenko, Leonid Kaluzhskiy, Tatsiana Tsybruk, Irina Grabovec, Maryia Kisel, Polina Shabunya, Natalia Soloveva, Nikita Vavilov, Andrei Gilep, Alexis Ivanov

**Affiliations:** 1Institute of Bioorganic Chemistry NASB, 5 Building 2, V.F. Kuprevich Street, 220141 Minsk, Belarus; tumilovicham@gmail.com (A.T.); tvshkel@gmail.com (T.T.); graboveci71@gmail.com (I.G.); marusen.kee@gmail.com (M.K.); polinashabunya@gmail.com (P.S.); agilep@yahoo.com (A.G.); 2Institute of Biomedical Chemistry, 10 Building 8, Pogodinskaya Street, 119121 Moscow, Russia; evgeniy.yablokov@ibmc.msk.ru (E.Y.); pavel79@inbox.ru (P.E.); gnedenko.oksana@gmail.com (O.G.); leonid.kaluzhskiy@ibmc.msk.ru (L.K.); n.shushkova@yandex.ru (N.S.); n.vavilov95@gmail.com (N.V.); alexei.ivanov@ibmc.msk.ru (A.I.); 3Research Centre for Medical Genetics, 1 Moskvorechye Street, 115522 Moscow, Russia; vika.basina12@gmail.com

**Keywords:** cytochrome P450, cytosolic sulfotransferase, cytochrome b5, surface plasmon resonance, enzymatic assay, in silico modeling, text mining, DHEA, steroidogenesis, monooxygenase system

## Abstract

Dehydroepiandrosterone (DHEA), a precursor of steroid sex hormones, is synthesized by steroid 17-alpha-hydroxylase/17,20-lyase (CYP17A1) with the participation of microsomal cytochrome b5 (CYB5A) and cytochrome P450 reductase (CPR), followed by sulfation by two cytosolic sulfotransferases, SULT1E1 and SULT2A1, for storage and transport to tissues in which its synthesis is not available. The involvement of CYP17A1 and SULTs in these successive reactions led us to consider the possible interaction of SULTs with DHEA-producing CYP17A1 and its redox partners. Text mining analysis, protein–protein network analysis, and gene co-expression analysis were performed to determine the relationships between SULTs and microsomal CYP isoforms. For the first time, using surface plasmon resonance, we detected interactions between CYP17A1 and SULT2A1 or SULT1E1. SULTs also interacted with CYB5A and CPR. The interaction parameters of SULT2A1/CYP17A1 and SULT2A1/CYB5A complexes seemed to be modulated by 3′-phosphoadenosine-5′-phosphosulfate (PAPS). Affinity purification, combined with mass spectrometry (AP-MS), allowed us to identify a spectrum of SULT1E1 potential protein partners, including CYB5A. We showed that the enzymatic activity of SULTs increased in the presence of only CYP17A1 or CYP17A1 and CYB5A mixture. The structures of CYP17A1/SULT1E1 and CYB5A/SULT1E1 complexes were predicted. Our data provide novel fundamental information about the organization of microsomal CYP-dependent macromolecular complexes.

## 1. Introduction

Dehydroepiandrosterone (DHEA), an important precursor of sex steroid hormones, is synthesized from cholesterol through a series of reactions catalyzed by cytochromes P450 (CYPs) and hydroxysteroid dehydrogenase mainly in the adrenal cortex (predominantly), gonads, and brain [1,2]. Notably, 75–90% of DHEA production comes from the adrenal cortex, and 10–25% of DHEA is produced in the testis or ovaries [1,2]. DHEA is produced via the Δ5-steroidogenic pathway in humans, in which CYPs play important roles (Figure 1). Thus, in mitochondria, the cholesterol side-chain cleavage enzyme (CYP11A1) converts cholesterol into pregnenolone [3]. Then, in the smooth endoplasmic reticulum (ER), steroid 17-alpha-hydroxylase/17,20 lyase (CYP17A1) hydroxylates pregnenolone at the C-17 position, followed by the cleavage of the bond between C-17 and C-20 to produce DHEA [4]. In this pathway, CYP17A1 works in conjunction with NADPH-dependent cytochrome P450 reductase (CPR) as an electron source and microsomal cytochrome b5 (CYB5A), increasing the activity of CYP17A1 by 10-fold as well as facilitating the binding of CYP17A1 to CPR [5,6,7].

In humans, almost all DHEA circulates in sulfonated form (DHEA-S), which is required for storage and delivery to peripheral tissues where DHEA synthesis is not available [8]. DHEA is sulfated by some cytosolic sulfotransferases (SULTs), in particular SULT2A1 and SULT1E1, exclusively in the adrenal glands [9,10]. SULTs catalyze the transfer of a sulfonate group from 3′-phosphoadenosine-5′-phosphosulfate (PAPS) to different substrates and are also involved in the metabolism of xenobiotics [11].

A molecular ‘machine’ is a well-known phenomenon based on the assembly of enzymes catalyzing successive steps of a metabolic pathway into a macromolecular protein complex. For example, this phenomenon is typical for enzymes of ribosome biogenesis [12], proteasomes [13], fatty acid synthase [14,15], pyruvate dehydrogenase [16], and others. Since CYP17A1 with the participation of its redox partners (CYB5A and CPR) and SULT1E1 or SULT2A1 are involved in two successive transformations of metabolites, we believe that these proteins can form a macromolecular complex. Thus, the aim of this work was to test the possibility of interactions of CYP17A1 and its redox partners (CYB5A and CPR) with SULT1E1 and SULT2A1 using bioinformatics, in silico modeling, and experimental methods (affinity purification combined with mass spectrometry (AP-MS), surface plasmon resonance (SPR) biosensor analysis, and enzymatic activity assessment).

## 2. Results

### 2.1. Relationships between SULTs and CYPs

To assess any relationships between SULTs and the microsomal P450-dependent monooxygenase system, the analysis of protein–protein interactions (PPIs) and gene co-expression data, as well as literature data, was performed. Using the STRINGdb, the PPI network between SULTs and cytochromes P450 was explored (Figure 2). There are four clusters of PPIs. The target cluster that included microsomal steroidogenic cytochromes P450 (CYP17A1, CYP19A1, and CYP21A2) and sulfotransferases (SULT2A1 and SULT1E1) and showed the highest interconnectivity compared to SULT1C2 and CYP3A43 (not connected with the above-mentioned CYPs and SULTs) was selected for further analysis (Figure 2).

The intracellular localization of SULTs and CYPs, according to the NextProt database (https://www.nextprot.org, accessed on 24 December 2022), is different because sulfotransferases are predominantly cytosolic enzymes, while cytochromes P450 are anchored through N-terminus in the endoplasmic reticulum membrane facing the globule to the cytoplasm side [17,18]. Taking into account that CYPs and SULTs are capable of forming heterogeneous complexes with noncytochrome proteins [19,20], we searched for interactomic information, but no evidence of any physical interactions between SULTs and CYPs was found.

The co-expression analysis of SULT and CYP genes showed that, firstly, there was a positive correlation in both normal and tumor tissues. It was found that in PCPG (pheochromocytoma/paraganglioma), there was a co-expression between *SULT1E1* and *CYP17A1* (*SULT1E1-CYP17A1*), *SULT1E1-CYP21A2, SULT2A1-CYP17A1*, and *SULT2A1-CYP21A2* genes, with Pearson correlation coefficients (R) equal to 0.60, 0.56, 0.81, and 0.73, respectively. Data on *SULT2A1-CYP17A1* gene co-expression were most fully obtained in brain tissues such as the amygdala (R = 0.51), frontal cortex (BA9) (R = 0.68), putamen (basal ganglia) (R = 0.57), spinal cord (cervical c-1) (R = 0.58), substantia nigra (R = 0.60), and caudate (basal ganglia) (R = 0.78), as well as in the ovary (R = 0.82) and testis (R = 0.60). The co-expression of *SULT2A1-CYP21A2* genes occurred in the cell cultures of transformed fibroblasts (R = 0.53) and EBV-transformed lymphocytes (R = 0.57). As for *SULT1E1-CYP19A1* and *SULT2A1-CYP19A1* genes, data on their co-expression were virtually absent. Figure 3 shows representative examples of the co-expression of *SULT2A1-CYP17A1* genes in PCPG and *CYP21A2-SULT1E1* genes in minor salivary glands with close values of Pearson or Spearman correlation coefficients. Thus, the gene co-expression patterns of SULTs and CYPs, which are also comparable with the literature data [21], likely provide evidence in favor of the existence of the corresponding heterogeneous PPIs.

Next, we explored the literature data hinting at interactions between SULTs and CYPs and analyzed the co-occurrence of terms in article texts. Text mining yielded 175 articles (Appendix A) in which the relationships between SULTs and CYPs were discussed. The abstracts and full texts of all the articles found (Appendix A) were further analyzed for evidence of any interactions (functional, co-expression, semantic, etc.) between cytochromes P450 and sulfotransferases, in particular between steroidogenic cytochromes P450 and cytosolic sulfotransferases. A visualization of MeSH keyword co-occurrence in these articles is shown in Appendix A. The map of co-occurrence contains four clusters with linked entities within each cluster and cross-linked with neighboring clusters. The largest number of links is typical for the following keywords: ‘sulfotransferases’, ‘CYP17A1 (two enzymatic activities)’, ‘aromatase (CYP19A1)’, ‘estrogens’, and ‘androgens’ located in different clusters. As can be seen in Appendix A, the biological context with the co-occurrence of CYPs and SULTs is mainly represented by transcriptomics and genomic studies (single nucleotide polymorphisms, gene expression, mRNA landscape profiling, the regulation of gene expression, and disease biomarkers). Other terms such as ‘progesterone reductase’, ‘sterol-sulfatase’, ‘catechol-O-methyltransferase’, and ‘estradiol dehydrogenase’ are also of interest because they are found in the texts, and, in fact, are related to the metabolic transformation of structurally similar metabolites and steroid hormones.

In summary, we could not find data on direct complexation and PPIs between sulfotransferases (SULT1E1 and SULT2A1) and steroidogenic microsomal cytochromes P450 (for example, CYP17A1, CYP21A2, and CYP19A1) in the target cluster shown in Figure 2; however, the results of article text mining as well as gene co-expression and co-occurrence analyses indicate the possibility of such interactions.

### 2.2. SPR Analysis of Interactions of CYP17A1 and Its Redox Partners with SULT1E1 and SULT2A1

We used SPR analysis to test the hypothesis of direct interactions of CYP17A1 and its redox partners (CPR and CYB5A) with SULT1E1 and SULT2A1. The typical SPR sensorgrams of binary interactions are shown in Figure 4. Table 1 shows the kinetic rate constants and equilibrium dissociation constant (Kd) of these interactions. The N-terminal hydrophobic domain-truncated CYP17A1 (CYP17A1tr) did not interact with immobilized SULTs. CYP17A1tr likely formed transient complexes with SULTs, which could not be detected by SPR biosensors due to their short lifetime. At the same time, full-length CYP17A1 (CYP17A1full) interacted with SULT1E1 and SULT2A1, and these interactions were characterized by Kd values in the submicromolar range (Table 1). It was found that SULTs had a high affinity for CYB5A. However, SULT2A1 interacted with CYB5A only in the presence of PAPS. The SULT2A1/CPR complexation was characterized by a Kd value in the micromolar range and detected only when CPR was immobilized on the optical chip.

### 2.3. Mutual Effects of CYPs and SULTs on Their Enzymatic Activities

To assess the functional significance of the detected PPIs, the mutual influence of the studied SULTs and CYP17A1 (as well as CYB5A) on each other’s enzymatic activity was evaluated. It should be noted that no effect of SULTs was observed on the activity of CYP17A1, regardless of the presence of PAPS. However, not everything is so unambiguous with the SULT activity. The activity of SULTs was evaluated in six different cases: in the absence of any CYP, in the presence of CYP17A1 (truncated and full-length forms), CYB5A, and simultaneously CYP17A1 (one of the forms) + CYB5A (Table 2). CYB5A by itself did not cause an increase in SULT activity but rather led to some decrease in it. However, CYB5A contributed to some increase in the activity of SULT1E1 and SULT2A1 in the presence of CYP17tr or CYP17A1full.

### 2.4. AP-MS Analysis

To confirm the hypothesis of any possible direct interaction of SULTs and the components of monooxygenase P450-dependent microsomal systems, we performed the AP-MS analysis of the spectrum of liver tissue lysate proteins, which can potentially interact with SULT1E1 and SULT2A1. The recombinant preparations of these enzymes are covalently immobilized on the inert sorbent (CNBr-Sepharose 4B) that allows it to specifically fish proteins out from liver tissue lysate. Because human SULTs have high amino acid sequence identity (about 70%) with rat orthologues, we isolated SULT protein partners from rat liver tissue lysate, which is particularly rich in SULT proteins and components of the cytochrome P450-dependent monooxygenase system [22].

Using SULT1E1 as a ‘bait’, 21 proteins were isolated from rat liver tissue lysate. Then, the seven most likely contaminant proteins (according to the CRAPome database statistics [23]) were excluded from this list, since they were present in more than 25% of AP-MS experiments. Among the remaining 14 proteins (Table 3 and Appendix A), CYB5A, Sult1b1, Sardh, Sult1a1, Sult1c1, and Ass1 were most reliably identified using two or more unique peptides. At the same time, only two proteins were identified as the potential protein partners of SULT2A1, but CYB5A was not among them. A detailed description of all the potential SULT protein partners identified is beyond the scope of this work.

### 2.5. In Silico Modeling of Protein Complexes

SPR analysis demonstrates direct interactions in the SULT/CYP17A1/CYB5A protein system; therefore, it is reasonable to predict the structures of corresponding molecular complexes. We used AlphaFold2 for the in silico modeling of four binary complexes (SULT1E1/CYP17A1, SULT2A1/CYP17A1, SULT1E1/CYB5A, and SULT2A1/CYB5A), taking into account both the monomeric and homodimeric forms of SULTs due to the lack of structural data on the oligomeric state of functionally active SULTs [24]. Apart from the built-in AlphaFold2 criteria (pLDDT and pTMscore) for evaluating the quality of the models, we also used the PDBePISA server to select the highest-priority models. Unfortunately, none of the protein–protein complexes involving SULT2A1 met the quality criteria. However, for SULT1E1, four models of proper quality were selected (Table 4 and Appendix A). The analysis of protein–protein interfaces in the complexes allowed us to find that all the involved amino acid residues of SULT1E1, CYP17A1, and CYB5A were outside the heme- and substrate-binding sites but were often in close proximity to such regions. The lists of amino acid residues forming the interfacial connections are provided in Appendix A. As for SULT1E1 monomers, the interface area of CYP17A1/SULT1E1 was 16% larger than that of CYB5A/SULT1E1; however, the percentage of the amino acid residues involved in protein–protein interface formation was almost three-fold higher for CYB5A (19.4%) vs. CYP17A1 (7.1%) at a constant value for SULT1E1 monomer (~10%). These findings are in good agreement with the estimated Kd value of CYB5A/SULT1E1 three-fold lower than that of CYP17A1/SULT1E1 (Table 1). It can be seen from Table 4 that CYP17A1/SULT1E1 (homodimer) has the largest interface area (1198 Å2), with almost two-fold number of H-bonds and salt bridges compared to CYP17A1/SULT1E1 (monomer) (836 Å2). Therefore, based on the model, we see that each SULT monomer is involved in the formation of a complex.

## 3. Discussion

Living systems are characterized by the co-localization of enzymes into multienzyme complexes [25] (such as proteasomes [13], fatty acid synthase [14,15], pyruvate dehydrogenase [16], etc.), which catalyze successive biochemical reactions. The last stage of prohormone DHEA biosynthesis and its subsequent sulfation are also successive biochemical reactions. We assumed that the enzymes (CYP17A1, CYB5A, CPR, SULT2A1, and SULT1E1) involved in these reactions can presumably participate in the formation of such multienzyme complexes. Therefore, we decided to test our assumption and evaluate the possibility of interaction between these enzymes and the mutual influence on their activity. Also, this assumption is of particular interest in light of the fact that microsomal CYPs (including CYP17A1) and cytosolic SULTs (including SULT2A1 and SULT1E1) participate in the first phase (the modification of lipophilic molecules) and second phase (conjugation) of xenobiotic transformation [26,27], respectively. Therefore, the very phenomenon that cytosolic SULTs and microsomal CYPs catalyze successive reactions is perhaps much broader than the example considered here. Thus, it can be assumed that there is a closer relationship between these enzymes. This may indicate new molecular mechanisms and the regulation of the formation of macromolecular complexes involving the components of CYP-dependent monooxygenase systems.

To perform a preliminary test of this assumption, we performed a bioinformatic analysis and literature search, which showed no data on direct interactions between CYPs and SULTs. Using STRINGdb, we found numerous associations between CYPs and SULTs, especially in the cluster of steroid-metabolizing microsomal CYPs (CYP17A1, CYP19A1, and CYP21A2) and DHEA-sulfating SULTs (SULT2A1 and SULT1E1). The co-occurrence of CYPs and SULTs terms was found in the article texts describing the biotransformation of xenobiotics, endogenous low-molecular-weight compounds, and steroid hormones, as well as the probable participation of proteins in neoplastic processes (Appendix A).

Another way to reveal the relationships between microsomal CYPs and cytosolic SULTs is to analyze gene co-expression data in both normal tissues and malignancies. The co-expression patterns of *SULT1E1* and *CYP17A1*, *SULT1E1* and *CYP21A2*, *SULT2A1* and *CYP17A1*, and *SULT2A1* and *CYP21A2* pairs of genes were found to have noticeable and high correlations (according to Chaddock scale) in pheochromocytoma–paraganglioma. It is known that the levels of corticosteroids and their metabolites in patients with pheochromocytoma–paraganglioma tend to increase, which is apparently associated with the accumulation of these hormones in tumor tissue [28]. The co-expressions of *SULT2A1* and *CYP17A1* as well as *SULT2A1* and *CYP21A2* pairs of genes were highly correlated in a number of normal tissues (brain, ovary, testes, fibroblasts, and lymphocytes). For *SULT1E1*, significant correlations were not found. It should be noted that some of these tissues are characterized by the synthesis of steroid hormones [29,30,31,32,33,34]. Thus, the significant co-expression of genes encoding steroidogenic microsomal CYPs and cytosolic SULTs in tissues producing different steroid hormones indicates molecular and, more likely, functional interactions.

Furthermore, we performed SPR analysis to confirm the direct interactions of CYP17A1 and its redox partners (CYB5A and CPR) with SULT2A1 and SULT1E1 immobilized on an optical chip. As a result, the possibility of a highly affine interaction of CYP17A1 and CYB5A with the studied SULTs was observed (Table 1). However, CPR could only interact with SULT2A1 and with less affinity than cytochromes. Additionally, a tendency of an increase in the association and dissociation rate constants of CYP17A1-SULTs complexes was observed in the presence of PAPS, albeit without significant changes in Kd values. This result is in line with the similar influence of low-molecular-weight substrates and products on the kinetic rate constants of the complexation of microsomal CYPs and their redox partners [35]. It should be noted that SULT2A1 interacted with CYB5A only in the presence of PAPS. To understand the nature of the effect of PAPS on this interaction, we need to understand how PAPS binds on the surface of SULT2A1. Information on the molecular mechanism of PAPS binding on the surface of mouse SULT1E1 was found in the literature. However, the common features identified in SULT1E1 for PAP (3′-phosphoadenosine-5′-phosphate) binding have also been noted in other SULTs [36]. PAPS has been described to bind with two conserved regions of mouse SULT1E1 (77% identity with human SULT1E1): amino acid residues 257–259 located at the beginning of GxxGxxK motif, as well as Arg130 and Ser138 [37]. Despite the fact that PAPS had no effect on the interaction of SULT1E1 with CYB5A, common features of PAPS binding with SULT1E1 appeared to be similar to those of other SULTs (including SULT2A1), suggesting that cytosolic and membrane-bound SULTs originate from a common ancestor [36]. Thus, if PAPS binds in a specialized site of SULT2A1, then it is possible for SULT2A1/CYB5A complexation to be driven by the allosteric effect of PAPS [38]. This effect of allosteric regulation with related changes in protein conformation may be involved in the interactions of other SULTs with CYPs [39]. Another possible mechanism underlying the interaction of SULTs and CYPs can be explained by considering PAPS as a ‘molecular glue’ [40], which agrees with our observation that the dissociation kinetics of the SULT1E1/CYB5A complex is slowed down in the presence of PAPS. However, the exact molecular mechanism of the PAPS effect in each individual case must be determined using structural analysis techniques.

The functional significance of novel PPIs with the participation of CYP17A1 and SULTs was verified in the biochemical experiments. The impact of CYB5A, which is known to affect the enzymatic activity of CYP17A1 [7,41], was also assessed. As can be seen from Table 2, CYP17A1full did not affect SULT2A1 enzymatic activity and had very little effect on the SULT1E1 enzymatic activity. This might be explained by the steric hindrance effects. Some microsomal CYPs are capable of forming oligomers with the involvement of the N-terminal domain of a protein in the oligomerization process [42]. CYP17A1 is predominantly in an oligomeric state, but it can exist in a monomeric state [43]. CYP17A1full is more likely to form oligomers in an aqueous solution because of its transmembrane domain. As a result, the binding sites of CYP17A1full become almost inaccessible for SULT1E1 and completely inaccessible for SULT2A1, which is likely associated with different positions of these binding sites on the surface of CYP17A1full for both SULTs. CYP17A1tr, due to the lack of its transmembrane domain, is presumably in a monomeric form, leading to a significant increase in SULT2A1 enzymatic activity. This observation was also confirmed in experiments with the truncated form, where CYP17A1 was in a monomeric state, and its effect on SULT1E1 activity did not change. Additionally, this difference in the effects of CYP17A1 on SULT1E1 and SULT2A1 can be explained by the different degree of their biochemical association with CYP17A1 [44].

CYB5A by itself decreased the enzymatic activity of both SULTs, but its combined effect with CYP17A1 led to an increase in the enzymatic activity of both SULTs in the case of CYP17A1tr and an increase in the enzymatic activity of only SULT1E1 in the case of CYP17A1full. After CYP17A1 interacts with CYB5A, the conformation of both proteins can change, as indicated by Yilin Liu et al. [41]. Presumably, these changes could eliminate the inhibitory effect of CYB5A on the activity of SULTs. Thus, our study’s SPR data on SULT/CYP17A1 and SULT/CYB5A interactions and biochemical data, as well as data on CYP17A1/CYB5A interaction reported in the literature [45,46,47], suggest the possible formation of a ternary complex SULT/CYP17A1/CYB5A. So, SPR analysis and biochemical experiments indicate that interactions of SULTs with CYP17A1 and redox partners can play structural and functional roles. We believe that such interactions can be valuable in terms of modulating SULT activity in vivo, as well as a functional coupling in the product–substrate axis, due to the spatial proximity of corresponding enzymes.

AP-MS was used to indirectly verify the interaction of the studied SULTs with the components of the cytochrome P450-dependent monooxygenase system in liver tissue lysate. CYB5A was absent in the list of the potential protein partners of SULT2A1 isolated from the PAPS-free media, whereas under the same conditions, CYB5A was identified in the list of potential SULT1E1 protein partners. Both of these facts are consistent with the results of the SPR analysis. Although the potential interaction of other identified proteins with the studied SULTs is of some interest, a detailed consideration of these results is beyond the scope of this work.

Having obtained data on the physical interaction of the studied SULTs with CYP17A1 and CYB5A, it seems interesting to evaluate the hypothetical structure of such protein–protein complexes. The models of the SULT/CYP17A1 and SULT/CYB5A complexes were predicted using the AlphaFold2 program, followed by its scoring assessment and the analysis of protein–protein interfaces using the PDBePISA web-based tool. Only SULT1E1 complexes with the studied cytochromes were successfully evaluated using PDBePISA and were selected for further analysis. Since it is not known whether dimers or monomers of SULT1E1 interact with these cytochromes, both types of complexes were generated. The analysis of interfaces reveals that hydrogen bonds and salt bridges seem to play a significant role. At the same time, the ratio of the interface area and the number of hydrogen bonds and salt bridges in the interfaces of the predicted models corresponds to the literature data providing information about the structure of protein–protein interfaces of known crystal structures of complexes [48,49]. No charge separation, which is typical for the interactions of CYPs/redox partners (the location of positive conditional charges on a CYP protein and negative ones on a redox partner), was observed [3,18,50,51]. ASN248, LYS250, and LYS257 of SULT1E1 contribute to the formation of the interface and stability of both types of complexes (with monomeric and dimeric SULT1E1) due to the formation of hydrogen bonds and salt bridges with cytochromes (Appendix A). LYS257 is known to form hydrogen bonds with ribose-3-phosphate of PAPS [52,53] that can likely modulate the kinetics of SULT1E1/CYB5A complexation, as can also be inferred from our SPR analysis. Thus, it is possible that the stabilization of the SULT-CYP-CYB5A protein system by PAPS provides optimal SULT performance.

The active site of CYPs is located in the depth of a protein globule [54,55]. Therefore, in order for the substrate to penetrate to the active center, in the structure of almost all cytochromes, there are channels for the entry and exit of the substrate [56]. CYP17A1 and SULT1E1 catalyze successive reactions, presumably being as a protein–protein complex. This led us to assume that the exit channel from the active site of CYP17A1 will neighbor the substrate entry to the SULT1E1 active site. To test this assumption in silico, channels in the structure of CYP17A1 were analyzed using the Caver analyst v program. 2.0 Beta. As a result, the expected orientation of the predicted channels was not observed (Appendix A). However, the close proximity of CYP17A1 and SULT1E1 molecules in the complex can be beneficial in the case of the isosteric regulation of SULT1E1 activity by CYP17A1 reaction products [57].

The existence of the identified multienzyme complex may be due to a biological consequence of the need to control the level of sex hormones. The sulfation of the hydroxyl group in the 3β-position of DHEA hinders its further conversion into a 3-keto steroid (androstenedione) catalyzed by 3β-hydroxysteroid dehydrogenase/delta 5->4 isomerase (HSD3B). The established patterns of multienzyme complexes suggest possible mechanisms for regulating the biosynthesis of androgens and estrogens; specifically, the presence of CYB5A in gonads decreases the formation of DHEA sulfate, thereby promoting the subsequent conversion of DHEA into androstenedione, while the controlled sulfation of DHEA by a multienzyme complex can prevent the hypersynthesis of androgens and estrogens, which can cause the development of a number of oncological processes.

## 4. Materials and Methods

### 4.1. Instruments

For the quality control of protein purification, MALDI-TOF Microflex LRF (Bruker Daltonik GmbH, Bremen, Germany) and program for spectra processing FlexAnalysis 3.0 (Bruker Daltonics GmbH, Germany) were used.

The HPLC analysis of enzymatic reaction products was performed on an Agilent Technologies 1200 Series chromatograph equipped with an Agilent 6410 Triple Quadrupole LC/MS system (Agilent, Santa Clara, CA, USA).

For the LC-MS/MS identification of proteins during the AP-MS procedure, a Q-Exactive HF-X mass spectrometer (Thermofisher Scientific, Waltham, MA, USA) was used.

Protein–protein interaction analysis was performed using SPR biosensors Biacore 3000 and T200 (Cytiva, Marlborough, MA, USA).

### 4.2. Protein Preparations

Highly purified (>95% according to SDS-PAGE) preparations of recombinant proteins (CYP17A1full, CYP17A1tr, CPR, CYB5A, SULT1E1, and SULT2A1) were carried out in the Institute of Bioorganic Chemistry of the National Academy of Sciences of Belarus using molecular cloning and heterologous expression in a bacterial system (*E. coli*) followed by metal affinity and ion exchange chromatography for protein purification. The protocols of CYP17A1full, CYP17A1tr, CPR, and CYB5A have been presented in our previous works [58,59,60]. The primary structures of the studied proteins are shown in Appendix A. MALDI-TOF spectra and SDS-PAGE images for the purity control of the proteins are presented in Appendix A.

SULT1E1 and SULT2A1 preparation methods are briefly described below. *E. coli* strains DH5α (containing a plasmid with the SULT1E1 gene) and Rosetta (containing a plasmid with the SULT2A1 gene) were used for heterologous expression. Bacterial strains were cultivated at 26 °C and 160 rpm for 48 h from the moment of induction using 0.5 mM IPTG (isopropyl-β-d-1-thiogalactopyranoside). The insoluble components of the mixture were removed via centrifugation at 22,000 rpm for one hour. Ni-NTA agarose was used for purification. The recombinant protein was eluted with 25 mM potassium phosphate buffer (KPB) (pH 7.4) containing 250 mM imidazole. Fractions were analyzed for protein content on a NanoDrop2000 spectrophotometer (Thermo Scientific, Waltham, MA, USA) at 280 nm and were applied to a hydroxyapatite chromatographic column for imidazole removal. The protein was eluted with a potassium phosphate buffer (600 mM, pH 7.4). The resulting fractions were analyzed using SDS-PAGE. Protein preparations were aliquoted and stored at −70 °C.

### 4.3. Reagents

Progesterone, NADPH, and 3′-phosphoadenosine-5′-phosphosulfate (PAPS) were obtained from Sigma-Aldrich (St. Louis, MI, USA). HBS-EP+ buffer (10 mM HEPES, 150 mM NaCl, 3 mM EDTA, 0.05% P20 detergent, pH 7.4), 10 mM sodium acetate buffers (pH 4.5, 5.0, and 5.5), a reagent kit for the covalent immobilization of proteins via primary amino groups (1-ethyl-3-(3-dimethylaminopropyl)-carbodiimide-HCl (EDC), and N-hydroxysuccinimide (NHS)) were obtained from Cytiva (Marlborough, MA, USA). The remaining reagents were obtained from local suppliers.

### 4.4. Bioinformatic Analysis

#### 4.4.1. Protein–Protein Interaction Network Construction

Construction of protein–protein interaction (PPI) networks, involving CYP1A1, CYP1A2, CYP1B1, CYP2A6, CYP2A7, CYP2A13, CYP2B6, CYP2C8, CYP2C9, CYP2C18, CYP2C19, CYP2D6, CYP2E1, CYP2F1, CYP2J2, CYP2R1, CYP2S1, CYP2U1, CYP2W1, CYP3A4, CYP3A5, CYP3A7, CYP3A43, CYP4A11, CYP4A22, CYP4B1, CYP4F2, CYP4F3, CYP4F8, CYP4F11, CYP4F12, CYP4F22, CYP4V2, CYP4X1, CYP4Z1, CYP5A1, CYP7A1, CYP7B1. CYP8A1, CYP8B1, CYP17A1, CYP19A1, CYP20A1, CYP21A2, CYP24A1, CYP26A1, CYP26B1, CYP26C1, CYP39A1, CYP46A1, CYP51A1, SULT1A1, SULT1A2, SULT1A3, SULT1A4, SULT1B1, SULT1C2, SULT1C3, SULT1C4, SULT1E1, SULT2A1, SULT2B1, SULT4A1, and SULT6B1, was performed using STRING database v.11.5 [61] (https://string-db.org, accessed 24 December 2022), with the following settings: meaning of network edges: evidence; active interaction sources: text mining, databases, co-expression, neighborhood, gene fusion, and co-occurrence; minimum required interaction score: (highest confidence 0.4), max number of interactors to show: query proteins only; k-means clustering.

#### 4.4.2. Co-Expression Analysis

The co-expression analysis of the genes encoding SULTs and CYPs (SULT1E1-CYP17A1, SULT1E1-CYP19A1, SULT1E1-CYP21A2, SULT2A1-CYP17A1, SULT2A1-CYP19A1, and SULT2A1-CYP21A2), in normal tissues and malignant neoplasms, was performed using the GEPIA2 portal [62] (http://gepia2.cancer-pku.cn/#correlation, accessed 27 December 2022). Pearson and Spearman correlation coefficient values R > 0.5, the number of clinical cases > 30, and *p*-value < 0.05 were set as cut-off criteria.

#### 4.4.3. Text Mining Analysis

The following online tools were used for articles text mining: Litsense [63], PubTator [64], the Coremine medical text mining tool (https://coremine.com/medical/, accessed 27 December 2022), and FACTA+ [65] (http://www.nactem.ac.uk/facta/cgi-bin/facta3, accessed 27 December 2022). The following keywords were used: “SULT1E1 and CYP17A1”, “SULT1E1 and CYP19A1”, “SULT1E1 and CYP21A2”, “SULT2A1 and CYP17A1”, “SULT2A1 and CYP19A1”, and “SULT2A1 and CYP21A2”. The search depth was not limited. Then, the target sample of 175 articles was exported from PubMed (https://pubmed.ncbi.nlm.nih.gov, accessed 10 January 2024).

The PIE resource [66] (http://www.ncbi.nlm.nih.gov/IRET/PIE/, accessed 27 December 2022) was also used to predict the probability of PPIs from the analysis of articles. This resource is archived and has not been updated since 8 May 2021.

The analysis of the co-occurrence of MeSH (Medical Subject Headings) keywords in the target sample of articles was necessary to extract the biomedical context that determined the spectrum of the associations between cytochromes P450 (CYP17A1, CYP19A1, and CYP21A2) and sulfotransferases (SULT1E1 and SULT2A1). The results of the co-occurrence analysis were visualized in VOSViewer (v. 1.6.19) [67]. The VOSViewer settings were the following: full counting method; minimum number of occurrences of a keyword—7 (of 712 keywords, 68 met the threshold; 44 keywords (after removing 24 keywords denoting gender, age groups, and experimental methods) were used for visualization); normalization method—association strength; advanced layout parameters—by default; clustering resolution—1.00; minimal cluster size—5; merge small clusters—yes; visualization weights—occurrences; lines minimal strength—4.

#### 4.4.4. Structural Prediction of Protein–Protein Complexes In Silico

The structural prediction of protein–protein complexes with unknown spatial structure was performed using AlphaFold2 [68] on ColabFold server v1.5.2 with default settings. Amino acid sequences of proteins were obtained from the UniProt database [69]. As a result, five hypotheses were generated for each case. The criteria for selecting the working hypotheses were the following: pLDDT (predicted local distance difference test) >70 [70] and pTMscore (predicted template modeling score) >0.5 [71]. pLDDT allows us to assess how well the polypeptide chain of a protein is folded, and pTMscore allows us to assess the quality of the protein–protein interface. However, we noticed that, in some cases, even visually unacceptable structures could have high pLDDT and pTMscore. Therefore, as an additional criterion for selection, we used the PDBePISA online resource [72] and focused on the contact area (>300 Å2) [73,74] and the number of salt bridges and hydrogen bonds between proteins in the models (preference was given to models with the largest number of connections). As a result, we selected several models of protein–protein complexes that met all the criteria. In addition, using PDBePISA, amino acids, which are capable of participating in the formation of a protein–protein interface and salt bridges or hydrogen bonds, were identified. The visualization of the models was performed in the UCSF Chimera 1.16 program [75].

Tunnels in the CYP17A1 structure were predicted using Caver analyst v. 2.0 with a probe diameter of 0.9 Å [76].

### 4.5. Surface Plasmon Resonance Analysis

Real-time measurements of PPIs were performed at 25 °C using CM5 optical chips coated with carboxymethyl dextran (CM-dextran). PPI sensorgrams were recorded as the change in the biosensor signal in resonance units (RUs) per unit of time (s). CPR, SULT2A1, and SULT1E1 were covalently immobilized onto the CM5 chip via amino groups. The carboxyl groups of CM-dextran were activated using a mixture of 0.4 M EDC and 0.1 M NHS for 7 min at a flow rate of 5 µL/min. Proteins (15 μg/mL) in the 10 mM sodium acetate buffer (pH 4.5 for CPR, pH 5.0 for SULT2A1, pH 5.5 for SULT1E1) were injected for 2 min at a flow rate of 5 μL/min. Next, the chip surface was washed with an HBS-EP+ buffer for 60 min.

CYP17A1full, CYP17A1tr, and CYB5A were injected as analytes in the HBS-EP+ buffer at concentrations from 50 to 5000 nM through the control (intact CM-dextran) and working (with immobilized protein) channels for 5 min at 15 µL/min. The dissociation of protein–protein complexes was recorded at the same flow rate for 5 min. In addition, the same experiments were carried out in the presence of PAPS (up to 1 mM) to assess the PPI-modulating effect. The resulting set of sensorgrams was obtained by subtracting the biosensor signal in the control channel from the signal in the working channel and analyzed according to the Langmuir binding model 1:1 using Biacore BIAevaluation software v. 4.1 (Cytiva, Marlborough, MA, USA).

The 1:1 (Langmuir) binding model [77] is a model for the 1:1 interaction between analyte (A) with immobilized protein (P) and is equivalent to the Langmuir isotherm for adsorption to a surface: A + P ↔ AP.
$$A+P\begin{array}{c} \mathrm{kon}\\ ↔\\ \mathrm{koff} \end{array}\mathrm{AP},$$
where koff is the dissociation rate constant, and kon is the association rate constant. The resulting equilibrium dissociation constant (Kd) is obtained as $\mathrm{Kd}=\frac{\mathrm{koff}}{\mathrm{kon}}$. 

### 4.6. Biochemical Tests

#### 4.6.1. CYP17A1 Enzymatic Activity in the Presence of SULTs

The enzymatic activity of CYP17A1 was assessed in the reduced system containing 0.5 μM of CYP17A1, 1 μM of CPR, and 50 μM of progesterone (P4) in a buffer comprising 10 mM of HEPES (pH 7.4) plus 10 mM of MgCl_2_. For estimating the effect of SULTs on CYP17A1 activity, 0.5 μM of SULT1E1 or SULT2A1 was added to the system (a molar ratio of CYP17A1:SULT was 1:1).

After incubating the protein aliquots at 37 °C for 10 min, the reaction was started by adding 0.5 mL of NADPH regeneration system (isocitrate, NADPH, isocitrate dehydrogenase). Then, the mixture was incubated at 37 °C for 30 min. The reaction was stopped by adding 5 mL of methylene chloride and subsequent centrifugation at 3000 rpm for 5 min. The upper organic layer was taken and evaporated in an argon atmosphere. After that, 100 µL of methanol was added to dissolve the precipitate. Product formation was detected with HPLC using a C18 Luna column (100 Å, 250 × 4.6 mm).

#### 4.6.2. SULT Enzymatic Activity in the Presence of CYP17A1 and CYB5A

SULT enzymatic activity was assessed in a reduced system containing 1 µM of SULT and 25 µM of PAPS in a 50 mM Tris-HCl buffer (pH 7.5) plus 10 mM of MgCl_2_. To test the effect of CYP17A1 on the catalytic activity of the studied SULTs, 1 μM of CYP17A1 with or without 1 μM of CYB5A was added to the system. (The molar ratio of CYP17A1:SULT:CYB5A was 1:1:1.) After incubating the protein aliquots at 37 °C for 10 min, the reaction was started by adding 50 μM of DHEA with subsequent incubation at 37 °C for 30 min. The reaction was stopped by adding 5 mL of methylene chloride and subsequent centrifugation at 3000 rpm for 5 min. The upper organic layer was taken and evaporated in an argon atmosphere. After that, 100 µL of acetonitrile was added to dissolve the precipitate.

Mass spectra for DHEA-S and DHEA standards were preliminarily obtained. The main ion with *m*/*z* 271.2– for DHEA-S was [M+H-sulfate-H_2_O]^+^, for DHEA, it was [M+H-H_2_O]^+^. DHEA-S was analyzed with LC-MS using a C18 Luna column (100 Å, 250 × 4.6 mm) and Agilent Technologies 1200 Series (Santa Clara, CA, USA).

### 4.7. Affinity Purification Combined with Mass Spectrometry

Liver tissue obtained from rats (Wistar line) was crushed and then washed with 150 mM NaCl at +4 °C. Homogenization was performed at 0 °C by grinding the tissue in a glass mortar with a pestle followed by processing in a SilentCrusher S homogenizer (Heidolph, Schwabach, Germany). Liver tissue lysate was prepared using a CelLytic™ MT Cell Lysis Reagent (Sigma-Aldrich, St. Louis, MO, USA) and protease inhibitor mix (Cytiva, Marlborough, MA, USA). Glycerol was added to the lysate up to 20% *v/v* for storage.

SULTs were covalently immobilized onto CNBr-Sepharose 4B (Cytiva, Marlborough, MA, USA). Sorbent preparation, the immobilization of a target protein, and the isolation of its protein partners from tissue lysate samples were described in our previous work [78]. HBS-P (10 mM HEPES, 150 mM NaCl, and 0.005% Tween-20, pH 7.4) was used as a working buffer for performing the AP-MS analysis of proteins. The proteins bound to the affinity sorbent were eluted through the double injection of 500 µL of 4% formic acid. CNBr-Sepharose 4B without protein immobilization was used as a control sorbent.

The eluted proteins were subjected to trypsin digestion according to the modified FASP protocol [79] for subsequent mass spectrometry (MS) identification. Briefly, 1 µL of the peptide mixture was loaded onto an Acclaim µ-Precolumn enrichment column (0.5 mm × 3 mm, particle size 5 µm, Thermo Scientific, Waltham, MA, USA) at a flow of 10 µL/min for 5 min in isocratic mode using 2% acetonitrile and 0.1% formic acid in deionized water as the mobile phase. Next, the peptides were separated on a Peaky™ C18 HPLC column (100 µm × 30 cm and 1.9 µm particle size) (Molecta, Moscow, Russia) in a gradient elution mode. The gradient was formed with mobile phase A (0.1% formic acid) and mobile phase B (80% acetonitrile and 0.1% formic acid) at a flow rate of 0.4 µL/min. The column was washed with 2% mobile phase B for 5 min, after which the concentration of mobile phase B was linearly increased to 40% for 75 min. Then, the concentration of phase B was linearly increased to 99% for 3 min, and after washing for 5 min, the concentration was linearly reduced to 2% for 2 min. The total duration of the analysis was 90 min.

MS analysis was performed in the positive ionization mode using a nanoelectrospray ionization (nESI) source (Thermo Scientific, Waltham, MA, USA). The following settings were used: an emitter voltage of 2.1 kV and a capillary temperature of 240 °C. Panoramic scanning was carried out in the mass range from 300 to 1500 *m*/*z* with a resolution of 120,000. In tandem scanning, the resolution was set to 15,000 in the mass range from 100 *m*/*z* to the upper limit, which was determined automatically based on the mass of the precursor. The isolation of precursor ions was carried out in a window of ±1 Da. The maximum number of ions allowed for isolation in the MS2 mode was set as ≤20 units, while the cut-off for selecting a precursor for tandem analysis was set as 400,000 units, and the normalized collision energy (NCE) was set as 29. For tandem scanning, only ions with charge states from 2+ to 4+ were taken into account. The maximum accumulation time was 50 ms for precursor ions and 40 ms for fragment ions. The automatic gain control (AGC) value for precursors and fragment ions was set to 106 and 105, respectively. All the measured precursors were dynamically excluded from the tandem MS/MS analysis for 40 s. Samples were analyzed in triplicate.

The analysis of the obtained MS data was performed using the MaxQuant v2.2.0.0 program [80]. Most of the program options were used by default. Methionine oxidation (M) and N-terminal acetylation (Protein N-term) were taken into account as variable modifications, and carbamidomethyl (C) was considered a fixed modification. The instrument type was set to Orbitrap with all the related presets by default. The database of protein records in FASTA format was downloaded from the UniProt [69] (download date: 27 December 2022). As a result, lists of the identified proteins were obtained for each target protein and control.

Further analysis of these lists was performed using the Perseus v2.0.7.0 program [81]. For each case, all proteins that occur in only one of the repeats were excluded from the final list. At the next stage, from the lists of the identified protein partners (obtained through the elution of proteins bound to the affine sorbent), all the proteins matching the control list (obtained through the elution of proteins bound to the control sorbent) were removed.

## 5. Conclusions

The cytochrome P450-dependent microsomal monooxygenase system performs extremely important functions in terms of the enzymatic transformation of exogenous compounds and the synthesis of many endogenous metabolites. Therefore, this enzymatic system cannot exist as a separate entity, which suggests interactions with its redox partners (CPR and CYB5A) and other cellular proteins. A metabolic cluster adapted for the implementation of the successive biochemical transformations of metabolites with the spatial proximity of enzymes is an optimal structural unit for the functioning of P450 cytochromes differing in substrate specificity. Proteins involved in this cluster can mutually regulate their functions. An example of such regulation is protein–protein interactions (PPIs) between different enzymes involved in successive stages of the biochemical transformation of compounds. PPIs provide the co-localization of enzymes, play a structural environment-forming role, and induce or inhibit the enzymatic activity of the interacting proteins. In our experiments, significant novel PPIs were discovered between the individual representative of microsomal cytochromes P450 (CYP17A1) and cytosolic sulfotransferases (SULT1E1 and SULT2A1). In addition, it was found that CYB5A and CPR can also interact with the studied SULTs. These interactions seem to be able to influence the functioning of the studied SULTs, which follows from our biochemical tests. Experiments on the affinity isolation of protein partners of the two SULTs from the liver tissue lysate also indicate the possibility of direct interaction between SULT1E1 and CYB5A. At the same time, the results of the study allow us to make an assumption about the triple complex CYP/SULT/CYB5A, which can be investigated in the future using special methods aimed at deciphering the structure of protein complexes. Given the data on the participation of CPR in interactions with all these proteins, the multiprotein complex may be more complicated than we assume. In particular, it may include proteins that we did not consider in this work. This idea can become a vector for further research.

## Figures and Tables

**Figure 1 ijms-25-02072-f001:**
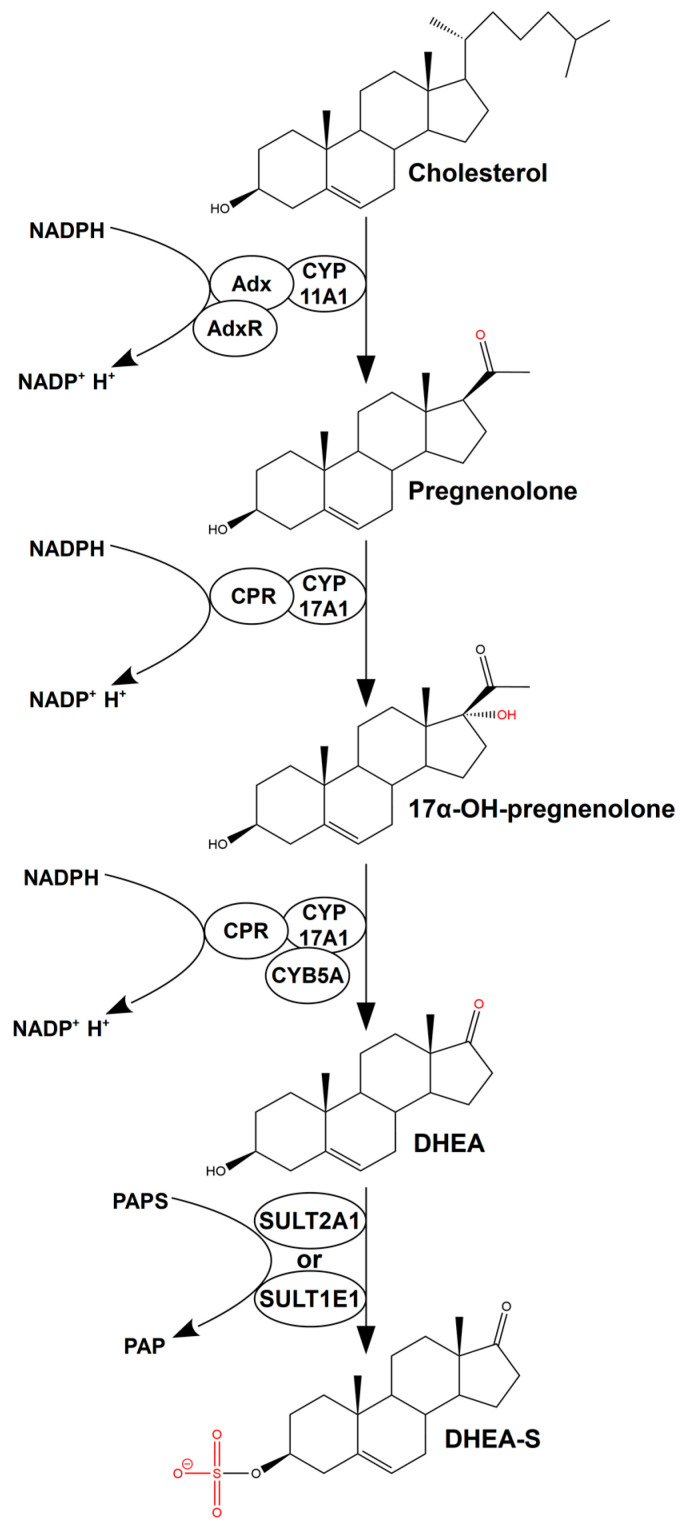
Biosynthesis of DHEA via the Δ5-steroidogenic pathway. Ovals denote metabolic enzymes (CYP11A1, CYP17A1, SULT1E1, and SULT2A1) and their redox partners (adrenodoxin—Adx, adrenodoxin reductase—AdR, NADPH-dependent cytochrome P450 reductase—CPR, and microsomal cytochrome b5—CYB5A) catalyzing through successive steps. The catalysis of reactions by CYPs and SULTs proceeds with the oxidation of NADPH and consumption of PAPS, respectively. Parts of the molecule modified during the reaction are highlighted in red.

**Figure 2 ijms-25-02072-f002:**
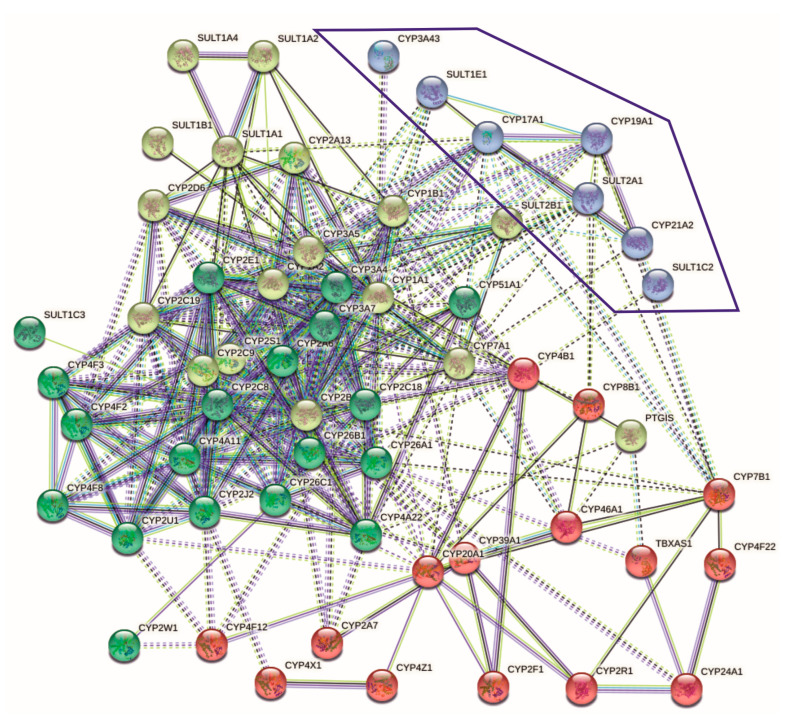
Clusterization of protein–protein interactions involving sulfotransferases (SULT family) and cytochromes P450 according to the STRINGdb data. The target cluster with the highest connectivity between SULTs and steroidogenic microsomal CYPs is highlighted in blue color. Red color—indicates the presence of fusion evidence, blue color—cooccurrence evidence, green color—neighborhood evidence, yellow color—textmining evidence.

**Figure 3 ijms-25-02072-f003:**
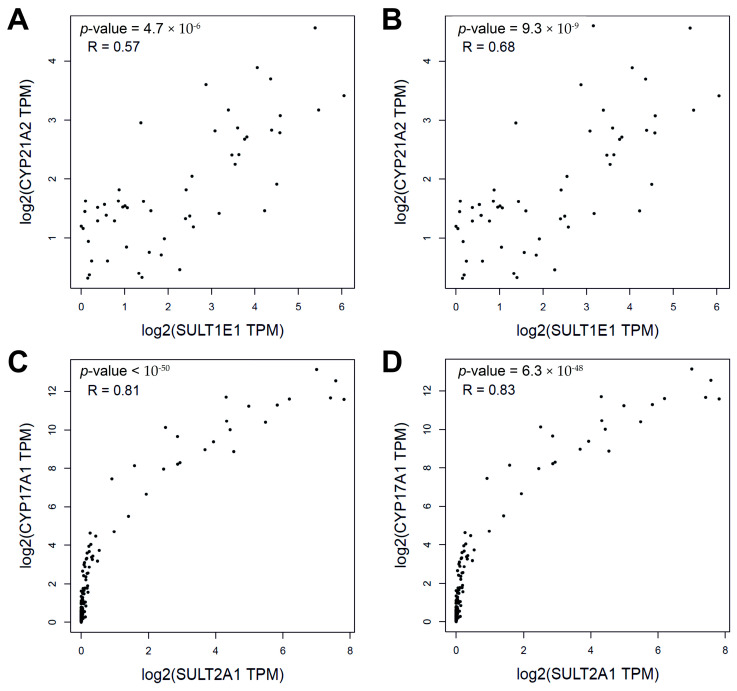
Examples of tissue-specific gene co-expression *SULT1E1-CYP21A2* in minor salivary glands: (**A**) Pearson correlation; (**B**) Spearman correlation; *SULT2A1-CYP17A1* in PCPG tumors: (**C**) Pearson correlation; (**D**) Spearman correlation. Gene co-expression was analyzed using the GEPIA2 web-based tool.

**Figure 4 ijms-25-02072-f004:**
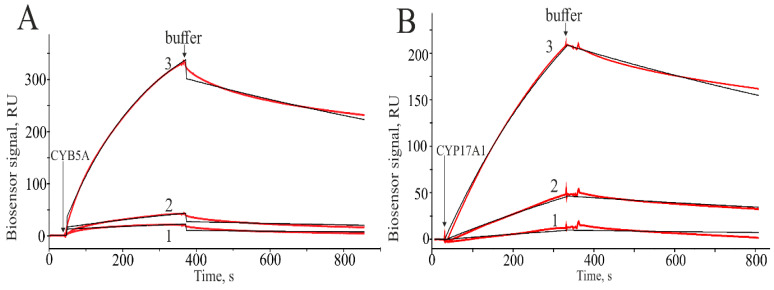
Sensorgrams of binding (red lines): (**A**) CYB5A at different concentrations 50 (1), 100 (2), and 500 nM (3) with immobilized SULT1E1 in the presence of 1 mM PAPS; fitting curves (theoretical models) are highlighted in black, χ^2^ = 4.90; (**B**) CYP17A1 at different concentrations 50 (1), 100 (2), and 500 nM (3) with immobilized SULT2A1; fitting curves (theoretical models) are highlighted in black, χ^2^ = 4.36.

**Table 1 ijms-25-02072-t001:** Values of kon (association rate constant), koff (dissociation rate constant), and Kd (equilibrium dissociation constant) for the interactions of full-length CYP17A1 and its redox partners with SULT1E1 and SULT2A1.

Interacting Proteins (Ligand/Analyte)	k_on_ (10^3^ × M^−1^ s^−1^)	k_off_ (10^−4^ × s^−1^)	K_d_ nM
SULT1E1/CYP17A1	2.96 ± 0.31	5.80 ± 0.49	196 ± 21
SULT1E1/CYP17A1 (PAPS *)	5.52 ± 0.52	7.60 ± 0.65	138 ± 15
SULT2A1/CYP17A1	4.45 ± 0.35	5.50 ± 0.50	124 ± 11
SULT2A1/CYP17A1 (PAPS)	13.10 ± 1.28	13.70 ± 1.50	105 ± 12
SULT1E1/CYB5A	13.80 ± 1.12	9.80 ± 1.00	71 ± 8
SULT1E1/CYB5A (PAPS)	7.65 ± 0.60	6.70 ± 0.50	88 ± 8
SULT2A1/CYB5A	n/d	n/d	n/d
SULT2A1/CYB5A (PAPS)	19.90 ± 2.20	14.00 ± 1.20	71 ± 7
CPR/SULT1E1	n/d	n/d	n/d
CPR/SULT1E1 (PAPS)	n/d	n/d	n/d
CPR/SULT2A1	0.57 ± 0.48	7.80 ± 0.80	1370 ± 126
CPR/SULT2A1 (PAPS)	0.67 ± 0.55	5.10 ± 0.40	957 ± 89

n/d—interaction is not detected; *—PAPS final concentration is equal to 1 mM.

**Table 2 ijms-25-02072-t002:** Enzymatic activity of SULTs in the presence of CYP17A1 and/or CYB5A.

	SULT1E1 Activity (min^−1^)	Resume	SULT2A1 Activity (min^−1^)	Resume
Control	0.09 ± 0.02	-	0.13 ± 0.06	-
CYP17A1tr + CYB5A	0.16 ± 0.03	↑ ↑ *	0.30 ± 0.07	↑ ↑
CYP17A1tr	0.08 ± 0.02	not changed	0.23 ± 0.02	↑ ↑
CYP17A1full + CYB5A	0.1700 ± 0.0009	↑ ↑	0.13 ± 0.08	not changed
CYP17A1full	0.120 ± 0.004	↑	0.13 ± 0.04	not changed
CYB5A	0.050 ± 0.005	↓	0.080 ± 0.006	↓

* ↑ ↑—increased; ↑—slightly increased; ↓—slightly decreased.

**Table 3 ijms-25-02072-t003:** A spectrum of potential partner proteins of SULT1E1 isolated from rat liver lysate.

Gene Names	Protein Names	Protein IDs	Unique Peptides (Unmodified)	Sequence Coverage, %
*Sult1b1*	Sulfotransferase family cytosolic 1B member 1	P52847	4	13.7
*Sardh*	Sarcosine dehydrogenase, mitochondrial	Q64380	4	3.2
*Sult1a1*	Sulfotransferase 1A1	P17988	3	10.3
*Sult1c1*	Sulfotransferase 1C1	P50237	3	18.4
*Cyb5a*	Cytochrome b5, microsomal	P00173	2	23.9
*Ass1*	Argininosuccinate synthase	P09034	2	4.9
*Aldob*	Fructose-bisphosphate aldolase B	P00884	1	5.5
*Rab7a*	Ras-related protein Rab-7a	P09527	1	6.8
*Cyp2d10 **	Cytochrome P450 2D10	P12939	1	2.2
*Cyp2d1 **	Cytochrome P450 2D1	P10633		
*Bdh1*	D-beta-hydroxybutyrate dehydrogenase, mitochondrial	P29147	1	5.8
*Pfkl*	ATP-dependent 6-phosphofructokinase, liver type	P30835	1	1.2
*Aldh3a2*	Fatty aldehyde dehydrogenase	P30839	1	1.9
*Gnb2 **	Guanine nucleotide-binding protein G(I)/G(S)/G(T) subunit beta-2	P54313	1	4.1
*Gnb4 **	Guanine nucleotide-binding protein subunit beta-4	O35353		
*Sult1c2a **	Sulfotransferase 1C2a	Q9WUW9	1	3.7
*Sult1c2 **	Sulfotransferase 1C2	Q9WUW8		

* The program cannot determine which of the two proteins is detected from one unique peptide, since this peptide is homologous for both proteins.

**Table 4 ijms-25-02072-t004:** AlphaFold quality criteria and the structural characteristics of the intermolecular interfaces of protein–protein complexes of SULT1E1 with CYP17A1 and CYB5A.

Complexes	AlphaFold2		PDBePISA		
	pLDDT	pTMscore	Interface Area (Å^2^)	Salt Bridges	H-Bonds
CYP17A1/SULT1E1 (monomer)	89.4	0.7	836	3	5
CYP17A1/SULT1E1 (homodimer)	90.6	0.6	1198	7	8
CYB5A/SULT1E1 (monomer)	88.1	0.8	699	6	7
CYB5A/SULT1E1 (homodimer)	90	0.8	702	0	4

## Data Availability

Datasets are available from the authors.

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
