# Peer review of "The Multienzyme Complex Nature of Dehydroepiandrosterone Sulfate Biosynthesis"

_ijms, 2024, doi:10.3390/ijms25042072_

Round 1

Reviewer 1 Report

Comments and Suggestions for Authors

Authors used surface plasmon resonance (SPR) to study the interactions between CYP17A1 and SULT2A1 or SULT1E1. SULTs also interacted with CYB5A and CPR. Interaction parameters of SULT2A1/CYP17A1 and SULT2A1/CYB5A complexes seem to be modulated by PAPS. In addition, authors used molecular modeling to decipher interactions between SULT1E1 and CYP17A1 successfully. Data from this study could be useful for further studies in this multi protein complex.

Comments:

SPR analysis: This reviewer is interested in knowing the effects of combinations of the following and their values.

1)      CYP17A1(PAPS)/CYB5A

2)      CYP17A1(PAPS)/CPR

3)      CYP17A1(PAPS)/SULT2A1 or SULT1E1/CYB5A

4)      CYP17A1(PAPS)/SULT2A1/CYB5A/CPR

PAPS: in the abstract section should be expanded. Line 81: PPI; expand.

Figure 1 should be revised with improved figure pixel quality.

Comments on the Quality of English Language

Moderate English corrections needed.

Author Response

Comments and Suggestions for Authors

Authors used surface plasmon resonance (SPR) to study the interactions between CYP17A1 and SULT2A1 or SULT1E1. SULTs also interacted with CYB5A and CPR. Interaction parameters of SULT2A1/CYP17A1 and SULT2A1/CYB5A complexes seem to be modulated by PAPS. In addition, authors used molecular modeling to decipher interactions between SULT1E1 and CYP17A1 successfully. Data from this study could be useful for further studies in this multi protein complex.

Comments:

SPR analysis: This reviewer is interested in knowing the effects of combinations of the following and their values.

1)                 CYP17A1(PAPS)/CYB5A

2)                 CYP17A1(PAPS)/CPR

3)                 CYP17A1(PAPS)/SULT2A1 or SULT1E1/CYB5A

4)                 CYP17A1(PAPS)/SULT2A1/CYB5A/CPR

 Authors’ response:

1 and 2 - We didn't have this idea previously because PAPS is not directly associated with CYPs. Thank you for giving us this idea, as it is particularly interesting in light of our findings implicating certain microsomal CYPs in processes associated with cytosolic sulfotransferases. The idea of ​​the influence of PAPS on the interaction of cytochromes with their redox partners can form the basis of our future work. However, interactions of cytochromes with their redox partners are beyond the focus of this article, which is devoted to the interaction of some cytosolic sulfotransferases with DHEA-producing CYP17A1.

3 and 4 - An SPR biosensor is designed primarily for the analysis of binary interactions. Interactions between three and more molecules, which are capable of interacting with each other, cannot be characterized quantitatively.

PAPS: in the abstract section should be expanded.

 Authors’ response:

Corrected accordingly. 

Line 81: PPI; expand. 

Authors’ response:

Corrected accordingly.

Figure 1 should be revised with improved figure pixel quality.

Authors’ response:

The original figure 1 with a resolution of 300 dpi was submitted to the editorial office. The loss of the figure quality is due to its insertion into the document.

Comments on the Quality of English Language

Moderate English corrections needed.

Authors’ response:

We have improved the quality of English in our manuscript.

Reviewer 2 Report

Comments and Suggestions for Authors

The paper delves into the intricate details of DHEA-S production, emphasizing its multienzyme complex formation. The MS is well-written in general. However, certain minor revisions are required to improve the clarity and precision of the paper:

1. Figure 3 should be revised to include an additional line and equation using R2. This will help readers better understand the correlation between the data points.

2. In Table 1, a more detailed explanation for K_on (association rate constant), K_off (dissociation rate constant), and K_d (dissociation constant) is needed to provide better insight into the reaction kinetics.

3. Table 3 contains information on protein-protein interactions obtained from PDBePISA, which suggests a relatively low interaction for salt bridges and H-bonds despite having a large interface area. The paper should further discuss what is the main interactions between these proteins.

Author Response

Reviewer 2

Comments and Suggestions for Authors

The paper delves into the intricate details of DHEA-S production, emphasizing its multienzyme complex formation. The MS is well-written in general. However, certain minor revisions are required to improve the clarity and precision of the paper:

  1. Figure 3 should be revised to include an additional line and equation using R2. This will help readers better understand the correlation between the data points.

Authors’ response:

The online tool GEPIA2, which we use to study gene co-expression, is not configured to obtain regression equations and highlight the trend line in the output graph. In figure 3, we have shown the original co-expression output from GEPIA2.

We did not intend to obtain any model based on gene co-expression, but rather extracted individual cases of gene co-expression between cytochromes P450 and sulfotransferases from the TCGA data array. This is of auxiliary importance for the formulation of a bioinformatic hypothesis about the physical interactions between cytochromes P450 and sulfotransferases and its further experimental verification.

There are online tools similar to GEPIA2 for studying gene co-expression, for example:

 https://gccri.bishop-lab.uthscsa.edu/shiny/correlation-analyzer/

 https://oncodb.org/correlation_expression.html

However, they do not allow to obtain regression equations and trend lines in the output data.

  1. In Table 1, a more detailed explanation for K_on (association rate constant), K_off (dissociation rate constant), and K_d (dissociation constant) is needed to provide better insight into the reaction kinetics.

Authors’ response:

Corrected accordingly. Explanations of the names of the constants have been added in the title of Table 1. An explanation of the meaning of the kinetic constants has been added to the Materials and Methods section (subsection 4.5).

  1. Table 3 contains information on protein-protein interactions obtained from PDBePISA, which suggests a relatively low interaction for salt bridges and H-bonds despite having a large interface area. The paper should further discuss what is the main interactions between these proteins.

Authors’ response:

The corresponding text has been added to the Discussion section:

“At the same time, the ratio of the interface area and the number of hydrogen bonds and salt bridges in the interfaces of the predicted models corresponds to the literature data, which summarize information about the structure of protein-protein interfaces of known crystal structures of complexes [DOI: 10.1093/protein/10.9.999 , DOI: 10.1006/jmbi.1998.2439].”

Reviewer 3 Report

Comments and Suggestions for Authors

 The manuscript was working on the synthetic enzymes of dehydroepiandrosterone and the interaction between the sulfotransferases and steroid hydroxylase and lyase. The authors used text mining, protein-protein networks, and gene co-expression analyses to indicate the relationships between SULTs and microsomal CYP isoforms. And surface plasmon resonance to detect the interactions between these proteins. The results showed that the enzymatic activity of sulfotransferases increases in the presence of only steroid lyase or hydroxylase. The work has some merits and deserves publication. However, some important data are missing in the manuscript. The detailed comments are listed below:

1)       Line 32, mix-->mixture.

2)       Line 32, add “The binding of” before “ CYP17A1/ SULT1E1 and CYB5A/SULT1E1”.

3)       Lines 71-92, rewrite “we expected 71 that these proteins function as part of a macromolecular protein complex”

4)       Lines 184-200, this is the only part of biochemistry, so the protein purification details including images and biochemical data should be given.

5)       And LC-MS/MS data should be added to the supplements.

Comments on the Quality of English Language

The English could be improved because some sentences are confusing.

Author Response

Reviewer 3

Comments and Suggestions for Authors

 The manuscript was working on the synthetic enzymes of dehydroepiandrosterone and the interaction between the sulfotransferases and steroid hydroxylase and lyase. The authors used text mining, protein-protein networks, and gene co-expression analyses to indicate the relationships between SULTs and microsomal CYP isoforms. And surface plasmon resonance to detect the interactions between these proteins. The results showed that the enzymatic activity of sulfotransferases increases in the presence of only steroid lyase or hydroxylase. The work has some merits and deserves publication. However, some important data are missing in the manuscript. The detailed comments are listed below:

1)       Line 32, mix-->mixture.

Authors’ response:

Corrected accordingly.

2)       Line 32, add “The binding of” before “ CYP17A1/ SULT1E1 and CYB5A/SULT1E1”.

Authors’ response:

Corrected accordingly. The sentence has been modified.

3)       Lines 71-92, rewrite “we expected 71 that these proteins function as part of a macromolecular protein complex”

Authors’ response:

Corrected accordingly. The sentence has been modified.

4)       Lines 184-200, this is the only part of biochemistry, so the protein purification details including images and biochemical data should be given.

Authors’ response:

Figure S6 has been added to the Supplementary file. It represents MALDI-TOF spectra and SDS-PAGE of purified proteins. As the biochemical characteristics of the enzymes are discussed in the article, the corresponding data obtained during the study on their enzymatic activities have already been presented in this article.

5)       And LC-MS/MS data should be added to the supplements.

Authors’ response:

Table S2 and Table 3 have been added to the supplementary file and main text, respectively.

Comments on the Quality of English Language

The English could be improved because some sentences are confusing.

Authors’ response:

We have improved the quality of English in our manuscript.

Reviewer 4 Report

Comments and Suggestions for Authors

In this article, Tumilovich and colleagues employ a mixture of bioinformatics and text data mining to imply likely protein-protein interactions specific cytochromes and sulfotransferases. They then perform surface plasmon resonance experiments to demonstrate these interactions in vitro, supplemented by affinity purification to indicate that such complexes likely also exist in vivo. The work is generally very well performed, and most of my comments are minor.

My major concern is lack of information on the literature mining. I understand the tools that the authors use to identify 175 potentially relevant articles on CYP-SULT interactions (lines 123-124), however it is not clear if the authors actually read these articles to ensure that no such prior interaction has been confirmed. These seems important as the authors stress the novelty of their finding.  A statement that all such articles were reviewed manually should be included, along with a list of the article titles in the supplementary material.

In section 2.2, a schematic diagram of the comparative domain structure of CYP17Afull and tr (probably 1-dimensional) would be very useful to explain its significance to the reader. Comparative diagrams of the SULTs would also be good.

In section 2.4, while I understand that the authors do not want to analyses all 14 putative protein binding partners of SULT1R1 in detail, I think it is important that the names of the interactors are included, ideally in a table which shows the confidence of each prediction.

The authors use AlphaFold2 for in silico modeling. Are solved structures also available in PDB for any or all of the proteins involved?

Comments on the Quality of English Language

The paper is generally well written, and fully comprehensible, although there are some grammar mistakes. Some example are:

Use of “human organism” instead of “humans” (lines 44, 53)

Use of “proper” instead of “required” (line 54)

“hinting on” instead of “hinting at” (line 122)

“Summarizing the abovementioned” when “In summary” is probably sufficient (line 138)

Author Response

Reviewer 4

Comments and Suggestions for Authors

In this article, Tumilovich and colleagues employ a mixture of bioinformatics and text data mining to imply likely protein-protein interactions specific cytochromes and sulfotransferases. They then perform surface plasmon resonance experiments to demonstrate these interactions in vitro, supplemented by affinity purification to indicate that such complexes likely also exist in vivo. The work is generally very well performed, and most of my comments are minor.

1) My major concern is lack of information on the literature mining. I understand the tools that the authors use to identify 175 potentially relevant articles on CYP-SULT interactions (lines 123-124), however it is not clear if the authors actually read these articles to ensure that no such prior interaction has been confirmed. These seems important as the authors stress the novelty of their finding.  A statement that all such articles were reviewed manually should be included, along with a list of the article titles in the supplementary material.

Authors’ response:

We have provided new Table S1 in the Supplementary  with a list of articles, whose abstracts and full texts have been manually analyzed with subsequent analysis of co-occurrence of biological terms.

A phrase has been added to the text: "Abstracts and full texts of all the found articles (Table S1) were further analyzed for the evidence of any (functional, co-expression, semantic, etc.) interactions between cytochromes P450 and sulfotransferases, in particular, between steroidogenic cytochromes P450 and cytosolic sulfotransferases."

2) In section 2.2, a schematic diagram of the comparative domain structure of CYP17Afull and tr (probably 1-dimensional) would be very useful to explain its significance to the reader. Comparative diagrams of the SULTs would also be good.

 Authors’ response:

Figure S5 has been added to the Supplementary file.

3) In section 2.4, while I understand that the authors do not want to analyses all 14 putative protein binding partners of SULT1R1 in detail, I think it is important that the names of the interactors are included, ideally in a table which shows the confidence of each prediction.

 Authors’ response:

Corrected accordingly. A Table 3 containing the names of 14 identified proteins has been added to the manuscript.

4) The authors use AlphaFold2 for in silico modeling. Are solved structures also available in PDB for any or all of the proteins involved?

Authors’ response:

The solved structures are available in PDB for most studied proteins. However, there is no crystal of full-length CYP17, whereas we were interested in modeling full-length protein complexes.

5) Comments on the Quality of English Language

The paper is generally well written, and fully comprehensible, although there are some grammar mistakes. Some example are:

Use of “human organism” instead of “humans” (lines 44, 53)

Authors’ response:

Corrected accordingly.

Use of “proper” instead of “required” (line 54)

Authors’ response:

Corrected accordingly.

“hinting on” instead of “hinting at” (line 122)

Authors’ response:

Corrected accordingly.

“Summarizing the abovementioned” when “In summary” is probably sufficient (line 138)

Authors’ response:

Corrected accordingly.

Round 2

Reviewer 3 Report

Comments and Suggestions for Authors

I have reviewed the revised manuscript, and all the issues have been revised and the manuscript could be accepted in the present form.